# Model-Based Dielectric Constant Estimation of Polymeric Nanocomposite

**DOI:** 10.3390/polym14061121

**Published:** 2022-03-11

**Authors:** Jiang Shao, Le Zhou, Yuqi Chen, Xue Liu, Mingbo Ji

**Affiliations:** 1The School of Mechanical Engineering, Jiangsu University of Science and Technology, Zhenjiang 212100, China; jiang-shao@hotmail.com; 2Research Department, Changzhou Betterial Film Technologies Co., Ltd., Changzhou 213000, China; zhoule@betterial.cn; 3School of Engineering and Applied Science, University of Pennsylvania, Philadelphia, PA 19104, USA; sevenchen9819@gmail.com; 4Henan Key Laboratory of Polyoxometalate Chemistry, College of Chemistry and Chemical Engineering, Henan University, Kaifeng 475004, China; 5Yantai Graduate School, Harbin Engineering University, Yantai 264000, China

**Keywords:** modeling, interphase, dielectric constant, nanocomposite, polymer

## Abstract

The interphase region widely exists in polymer-based nanocomposites, which affects the dielectric properties of the nanocomposites. General models, such as the Knott model, are often used to predict the dielectric constant of nanocomposites, while the model does not take the existence of interphase into account, which leads to a large deviation between the predicted results and the experimental values. In this study, a developed Knott model is proposed by introducing the interphase region and appropriately assuming the properties of the interphase. The modeling results based on the developed model are in good agreement with the experimental data, which verifies the high accuracy of the development model. The influence of nanoparticle loading on the effective volume fraction is further studied. In addition, the effects of the polymer matrix, nanoparticles, interphase dielectric and thickness, nanoparticle size and volume fraction on the dielectric properties of the nanocomposites are also investigated. The results show that polymer matrix or nanoparticles with a high dielectric and thick interphase can effectively improve the dielectric properties of the materials. Small size nanoparticles with high concentrations are more conducive to improving the dielectric properties of the nanocomposites. This study demonstrates that the interphase properties have an important impact on the dielectric properties of nanocomposites, and the developed model is helpful to accurately predict the dielectric constant of polymer-based nanocomposites.

## 1. Introduction

Large capacity energy storage technology plays a vital role in the application fields of smart grid construction [1], new energy generation [2] and electric vehicles [3]. Compared with battery and supercapacitor storage, high energy storage electrolyte capacitors have many advantages in safety, economic cost and charge-discharge rate [4,5]. At present, electrostatic capacitors have high power density due to their fast charge-discharge ability among the available electric energy storage devices, while the low energy density limits their applications [6,7]. Therefore, much research has been focusing on how to improve the performance of energy storage materials, especially dielectric energy storage materials [8,9]. Ceramic dielectric nanoparticles, such as barium titanate (BT) [10], lead zirconate titanate (PZT) [11], barium strontium titanate (BST) [12] and copper calcium titanate (CCTO) [13], which have a high dielectric constant and excellent anti-aging properties [14,15,16], are one of the most common used dielectric materials in energy storge. However, there are several shortcomings that limit the application of ceramic dielectric nanoparticles, for instance, the high sintering temperature, complex preparation process, poor flexibility, high dielectric loss and low electrical strength [17,18,19]. Polymeric materials, such as polyethylene, polypropylene and epoxy resin, have attracted much attention in their application in dielectric energy storage materials due to their excellent mechanical properties, high breakdown strength and low dielectric [20,21], while the relative dielectric constant of most polymeric materials is usually low [22]. Therefore, it is difficult to obtain excellent energy storage devices from a single dielectric material. One of the popular methods to overcome this disadvantage is to mix ceramic nanoparticles into the dielectric polymer matrix [23]. Blending is the most commonly used method for preparing polymer-based nanocomposites, which directly disperses inorganic nanoparticles in the polymer matrix. Inorganic nanoparticles, such as SiO_2_, CaCO_3_, TiO_2_, Al_2_O_3_ and Fe_3_O_4_ are often used to prepare nanocomposites.

Many works have been conducted on polymer-based dielectric materials and excellent results have been obtained [24,25,26]. For example, BT material has the advantages of high dielectric constant and low dielectric loss, which is the most commonly used ceramic filling particle at present [27]. It is worth noting that although multi-factor experiments can obtain polymer-based materials with good dielectric properties, most of the experimental processes are repetitive and redundant, which not only takes time but also increases the economic cost. Mathematical modeling is a simple and effective method to deepen the understanding of experimental mechanisms [28]. By constructing an appropriate theoretical model, the dielectric properties of polymer-based nanocomposites can be accurately predicted. Several good theoretical models have been reported, such as the Bruggeman model [29], which was proposed to estimate the dielectric constant of nanocomposites that contain a high concentration of particles. By regarding the filled particles as homogeneous spheres dispersed in a continuous polymer matrix, the Rayleigh model [30] was also proposed to predict the dielectric constant of nanocomposites, which greatly reduced the time required for the experiment. Other models that need complex equations and expensive software were also built to predict the properties of nanocomposites, such as the Monte-Carlo simulation [31] and representative volume element model [32].

Although the models mentioned above were well established, the authors simply investigated the effect of additive components on the properties of the nanocomposites, while the interaction between additive particles and polymer matrix was ignored. In fact, in a micro-environment, there is an interphase formed by the interaction between additive particles and polymer matrix, which significantly affects the properties of the nanocomposites, and several works have reported that the existence of the interphase affects the mechanical and electrical properties of composite materials [33,34]. It can also be expected that the interphase has effects on the overall dielectric constant of nanocomposites. The additive particle surrounded by polymer matrix can be regarded as an equivalent sphere with a core-shell structure. Therefore, by parameter simulation with the interphase property, the dielectric constant of the nanocomposite can be accurately predicted. Similarly, the dielectric constant from modeling results will be underestimated if the interphase is ignored.

In the study of plastic foam, Knott et al. [35] introduced a model in which filler particles were regarded as small cubes surrounded by substrates of the same thickness, and the effective capacitance of the foam (the effective permittivity) could be simulated by using the capacitance of the cube and the substrate. However, they also neglected the existence of the interphase, which limits the accuracy of the equation. In this study, the effect of interphase on the dielectric properties of nanocomposites is investigated. By extending the Knott equation, the effects of a polymer matrix, particles, interphase dielectric, interphase thickness, particle size and volume fraction on the dielectric properties of nanocomposites are studied. The developed model provides a promising method for predicting the dielectric properties of nanocomposites.

## 2. Model Description

The Knott model was developed to give an engineering estimation of the dielectric constant of plastic foams, which consists of a polymer base and gas. The gas in the foam was assumed to be a small cubic lattice, covered by a thin polymer base. The elemental unit is composed of a gas cubic lattice and polymer cover, and it was placed in a uniform electric field. Based on this assumption, the Knott model was proposed as [35,36]:(1)$$\varepsilon =\varepsilon_{m}-\varepsilon_{m}\frac{\left(\varepsilon_{m}-\varepsilon_{f}\right)V_{f}}{\varepsilon_{f}+\left(\varepsilon_{m}-\varepsilon_{f}\right)V_{f}{}^{\frac{1}{3}}}$$
where $\varepsilon_{f}$ and $\varepsilon_{m}$ are the dielectric permittivity of gas and polymer base, respectively. $V_{f}$ is the volume fraction of gas.

Although this model was designed for plastic foams, it can also be extended to estimate the dielectric permittivity of other materials. Considering the shape of a small cubic lattice, this Knott model is applied for predicting the dielectric constant of polymeric composites containing spherical particles. Even though a spherical particle is slightly different from a cubic lattice, it has been demonstrated that this small difference in the shape of fillers has a minor influence on the dielectric permittivity of nanocomposites.

However, as we have stated previously, the Knott model did not take into consideration the interphase region. Besides, the present Knott model fails to include the effect of particle parameters. To introduce them, the development of the Knott model is necessary.

Tanaka et al. [37] proposed a simple formula that can be used to estimate the dielectric constant of the hybrid particle and the hybrid particle has a configuration of the core-shell structure, as illustrated in Figure 1. The Tanaka formula is expressed as:(2)$$\varepsilon_{h}=\varepsilon_{s}\frac{\varepsilon_{c}\left(1+2\rho \right)+2\varepsilon_{s}\left(1-\rho \right)}{\varepsilon_{c}\left(1-\rho \right)+\varepsilon_{s}\left(2+\rho \right)}$$
where $\varepsilon_{h}$ is the effective permittivity of hybrid particles. $\varepsilon_{c}$ and $\varepsilon_{s}$ are dielectric permittivity of core part and shell part of hybrid particle, respectively. ρ is a parameter that defined by the size of the core part and shell part:(3)$$\rho =\frac{R_{c}{}^{3}}{{\left(R_{c}+R_{s}\right)}^{3}}$$
where $R_{c}$ is the radius of core filler and $R_{s}$ is the thickness of the shell part.

In this study, the Tanaka formula is utilized to introduce the effect of the interphase region. The nanoparticle is considered as the core part of the hybrid particle while the surrounding interphase region is treated as the shell part of the hybrid particle. As a result, the dielectric permittivity $\varepsilon_{\mathrm{ef}}$ of the effective particle, composed of filler and surrounding interphase, can be expressed as:(4)$$\varepsilon_{\mathrm{ef}}=\varepsilon_{i}\frac{\varepsilon_{f}\left(1+2\rho \right)+2\varepsilon_{i}\left(1-\rho \right)}{\varepsilon_{f}\left(1-\rho \right)+\varepsilon_{i}\left(2+\rho \right)}$$
and
(5)$$\rho =\frac{R^{3}}{{\left(R+R_{i}\right)}^{3}}$$
where $\varepsilon_{f}$ and $\varepsilon_{i}$ are dielectric permittivity of filler and surrounding interphase, respectively. R denotes the radius of inclusions and $R_{i}$ represents the thickness of the interphase.

Since the volume fraction of nanoparticles is calculated by the following expression:(6)$$V_{f}=\frac{V_{p}}{V_{c}}$$
where $V_{p}$ represents the volume of nanoparticles and $V_{c}$ is the volume of the nanocomposite. The volume of spherical inclusions $V_{p}$ is simply calculated as:(7)$$V_{p}=\frac{4}{3}{\pi R}^{3}$$

Therefore, the volume of the effective particle can be derived as:(8)$$V_{\mathrm{ep}}=\frac{4}{3}\pi {\left(R+R_{i}\right)}^{3}$$

Based on Equations (6)–(8), the volume fraction $V_{\mathrm{ef}}$ of the effective particle is derived:(9)$$V_{\mathrm{ef}}=V_{f}{\left(\frac{R+R_{i}}{R}\right)}^{3}$$

As a result, a developed Knott model that considers the parameters of filled particles as well as the interphase properties is developed as:(10)$$\varepsilon =\varepsilon_{m}-\varepsilon_{m}\frac{\left(\varepsilon_{m}-\varepsilon_{\mathrm{ef}}\right)V_{\mathrm{ef}}}{\varepsilon_{\mathrm{ef}}+\left(\varepsilon_{m}-\varepsilon_{\mathrm{ef}}\right)V_{\mathrm{ef}}{}^{\frac{1}{3}}}$$

## 3. Results and Discussion

In order to verify the applicability of the developed Knott model, it is used to predict the dielectric constant of polymer-based nanocomposites, in which the parameters are obtained from the published literature. The prediction results are compared with the experimental data to examine the accuracy of the developed model. Figure 2 displays the experimental and modeling results of the dielectric constant of three nanocomposites under different particle loads. It can be seen that the dielectric constant calculated by the developed model is very consistent with the data obtained from the actual experiment while a much larger deviation between the modeling results of the original Knott model and experimental results can be observed, which indicates that the developed model can accurately predict the dielectric constant of nanocomposites and can be used for the pre-evaluation of dielectric materials. In fact, the developed Knott model has widespread applicability in different nanocomposites. The experimental data of multiple samples have been utilized to validate the developed Knott model, as shown in Appendix A. Introducing the interphase into the Knott model is responsible for the accurate simulation results. By reasonably assuming the properties of the interphase and optimizing the calculation parameters, the accurate dielectric constant of the nanocomposite can be obtained, which will help to reduce the experimental steps to save time and cost.

The dielectric properties of nanocomposites are affected by many factors, including particle volume fraction, dielectric properties of polymer matrix and nanoparticles, thickness and dielectric properties of interphase. The modeling results of particle load and particle volume fraction are displayed in Figure 3. Obviously, for the three selected nanocomposites, the simulated effective volume fraction is higher than the theoretical particle volume fraction. As shown in Figure 3b, there is a small difference between the modeling results and the experimental data when the particle loading is low. However, as the particle load increases gradually, the deviation between theoretical and modeling results increases and the effective particle volume fraction is much higher than the theoretical data, especially for Figure 3a,c. The existence of the interphase is responsible for these results, which effectively expands the action range of particles, thus increasing their effective volume fraction. Therefore, the actual effect of particles will be largely underestimated if the interphase is ignored.

The nanocomposite is composed of polymer matrix and nanoparticles; therefore, the properties of polymer matrix and particles jointly determine the dielectric constant of the nanocomposite [41,42]. To further investigate the effect of interphase on the dielectric constant of the nanocomposite, the dielectric constant of nanocomposite εc was simulated with the dielectric constant of the polymer matrix ($\varepsilon_{m}$) and particles ($\varepsilon_{f}$) as R_i_ = 10 nm, R = 100 nm, V_f_ = 0.1 and $\varepsilon_{i}$ = 20. As shown in Figure 4, it is obvious that the maximum dielectric constant of the nanocomposite $\varepsilon_{c}$ is obtained at the highest $\varepsilon_{m}$ and $\varepsilon_{f}$. As the dielectric constant of the polymer matrix is low, for instance, $\varepsilon_{m}$ is less than 3.5, the dielectric constant of the nanocomposite $\varepsilon_{c}$ is determined by the properties of the polymer matrix, and no matter how many particles are added, indicating a low $\varepsilon_{m}$ is sufficient for the nanocomposites to have a low dielectric constant. With the increase of the dielectric properties of a polymer matrix, the dielectric constant of nanocomposites increases, and $\varepsilon_{c}$ is determined by both the dielectric properties of polymer matrix and particles. High dielectric constant particles significantly improve the dielectric properties of the nanocomposites. Therefore, in the actual experimental process, it is a feasible method to select particles with high dielectric constant to prepare nanocomposites with high dielectric properties.

In order to improve the dielectric constant of particles, some surface modification can be performed. The compatibility between nanoparticles and polymer matrices can be improved by introducing active functional groups through surface chemical or physical modification. For instance, Kim et al. [36] improved the compatibility between polymer matrix and particles by coating different organic phosphoric acids on the surface of barium titanate particles. Zhang et al. [43] chemically integrated the third monomer chlorofluoroethylene (CFE) or chlorotrifluoroethylene (CTFE) into PVDF-trifluoroethylene to form a ternary mixture, and they found that the ternary polymer is an electrical ferroelectric relaxation material with excellent dielectric properties.

The effects of interphase thickness $R_{i}$ and dielectric constant $\varepsilon_{i}$ on the dielectric properties of the polymer-based nanocomposites are also investigated as $\varepsilon_{f}$ = 50, $\varepsilon_{m}$ = 6, R = 100 nm and $V_{f}$ = 0.1. As shown in Figure 5, the polymer-based nanocomposites obtain the highest dielectric constant at the maximum $R_{i}$ and $\varepsilon_{i}$, and the lowest dielectric properties are obtained at the minimum $R_{i}$ and $\varepsilon_{i}$. However, the dielectric constant of the polymer-based nanocomposites ($\varepsilon_{c}$) is determined only by the interphase thickness $R_{i}$ as the interphase is thin. It can be seen that the dielectric properties of the polymer-based nanocomposites are determined by both $R_{i}$ and $\varepsilon_{i}$ as the increase of interphase thickness and dielectric constant. It should be noted that the interphase thickness plays a more important role than the interphase dielectric constant, in which thin thickness always results in low dielectric properties of the nanocomposite. Therefore, a thick and high dielectric interphase is helpful to improve the dielectric properties of nanocomposites. By evenly dispersing the particles in the polymer matrix to increase the contact area, a thick interphase may be formed.

Particle radius R and volume fraction $V_{f}$ are also important factors affecting the dielectric properties of polymer-based nanocomposites. As shown in Figure 6, the highest nanocomposite dielectric constant is obtained at the smallest R with the highest $V_{f}$ as $\varepsilon_{f}$ = 50, $\varepsilon_{m}$ = 6, $\varepsilon_{i}$ = 20, $R_{i}$ = 25 nm and the lowest dielectric constant of the polymer-based nanocomposite is obtained in the particles with a maximum size and low volume fraction. Although the particle radius and volume fraction both determine the dielectric properties of the polymer-based nanocomposites, the volume fraction plays more role than particle size, in which low volume fraction results in the low dielectric constant of nanocomposite no matter what size particles are added. It should be noted that the large particle size seriously damages the overall properties of the nanocomposites. Therefore, filling small size particles with high concentrations is an effective method to improve the dielectric properties of nanocomposites. The aggregation of nanoparticles easily occurs, as the small size of the filled particles with high concentrations. Thus, it is necessary to modify or evenly disperse these particles into a polymer matrix. Surfactants or dispersants are commonly applied to improve the surface of nanoparticles [44,45], resulting in uniformly dispersed particles in a polymer matrix. For instance, Kim et al. [46] modified BT particles to prepare nanocomposite, and the obtained thin film materials exhibit good dispersibility and high dielectric strength. In addition, process methods, such as magnetic stirring [29] and ultrasonic dispersion [47] are always helpful in enhancing the dispersive property of particles into a polymer matrix.

## 4. Conclusions

In this study, a developed Knott model is proposed by introducing the interphase properties of nanoparticles. By comparing experimental data with modeling results, it is found that the developed model can accurately predict the dielectric properties of polymer-based nanocomposites. The effect of particle loading on the effective volume fraction is further studied. It demonstrates that the effective volume fraction is close to the theoretical data as the particle concentration is low, while the effective volume fraction is much higher than the theoretical data as the loading concentration is large. The existence of the interphase effectively expands the action range of the filled particles. Furthermore, the effects of a polymer matrix, particles, interfacial dielectric and thickness, particle size and volume fraction on the dielectric properties of the nanocomposites were investigated and the results show that high dielectric polymer matrix, particles and thick interphase can effectively improve the dielectric properties of the materials. High concentration and small size particles are more conducive to enhancing the dielectric properties of the nanocomposites. This study demonstrates that the interphase properties have an important impact on the dielectric properties of nanocomposites, and the developed model is promising to accurately predict the dielectric constant of polymer-based nanocomposites.

## Figures and Tables

**Figure 1 polymers-14-01121-f001:**
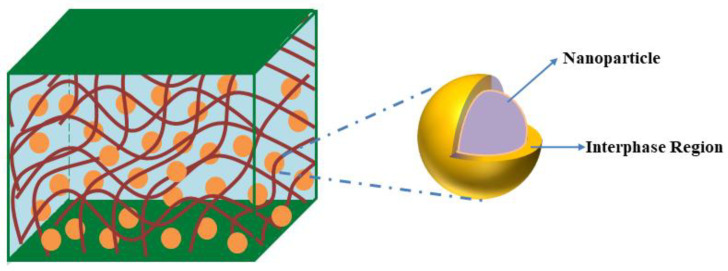
A schematic illustration of the hybrid particle in a nanocomposite.

**Figure 2 polymers-14-01121-f002:**
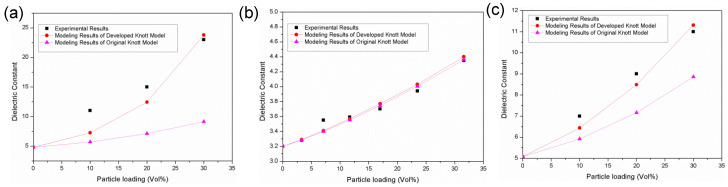
The experimental results and the modeling results of the dielectric constant of polymer-based nanocomposites (**a**) BaTiO_3_-epoxy resin [38], (**b**) Al_2_O_3_-epoxy resin [39], (**c**) BaTiO_3_-epoxy resin [40].

**Figure 3 polymers-14-01121-f003:**
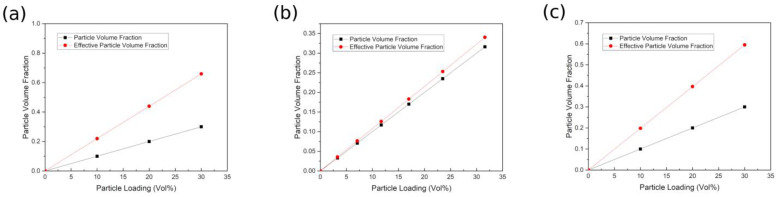
Comparison of particle volume fraction and effective particle volume fraction of polymer-based nanocomposites (**a**) BaTiO_3_-epoxy resin (**b**) Al_2_O_3_-epoxy resin (**c**) BaTiO_3_-epoxy resin.

**Figure 4 polymers-14-01121-f004:**
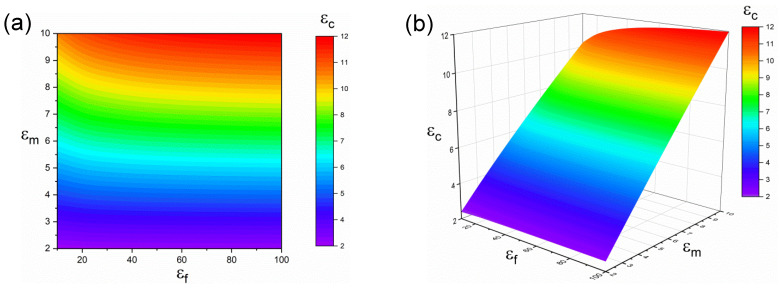
The dependence of $\varepsilon_{c}$ to the $\varepsilon_{m}$ and $\varepsilon_{f}$ when $R_{i}$ = 10 nm, R = 100 nm, $V_{f}$ = 0.1 and $\varepsilon_{i}$ = 20 (**a**) contour plot; (**b**) 3D plot.

**Figure 5 polymers-14-01121-f005:**
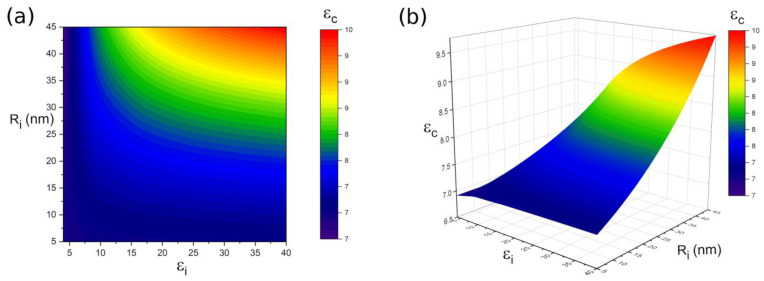
The dependence of $\varepsilon_{c}$ to the $\varepsilon_{i}$ and $R_{i}$ when $\varepsilon_{f}$ = 50, $\varepsilon_{m}$ = 6, R = 100 nm and $V_{f}$ = 0.1 (**a**) contour plot; (**b**) 3D plot.

**Figure 6 polymers-14-01121-f006:**
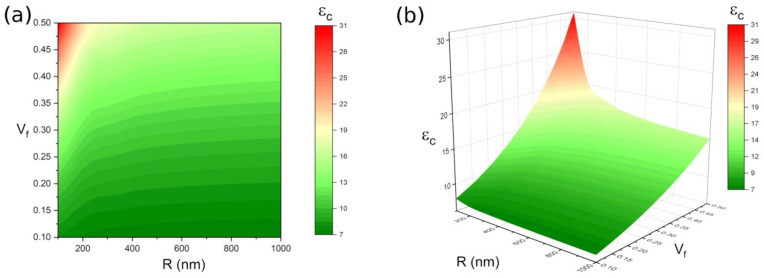
The dependence of $\varepsilon_{c}$ to the R and $V_{f}$ when $\varepsilon_{f}$ = 50, $\varepsilon_{m}$ = 6, $\varepsilon_{i}$ = 20, $R_{i}$ = 25 nm (**a**) contour plot; (**b**) 3D plot.

## Data Availability

Not applicable.

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
