# Peer review of "Model-Based Dielectric Constant Estimation of Polymeric Nanocomposite"

_polymers, 2022, doi:10.3390/polym14061121_

Round 1

Reviewer 1 Report

Authors present a study on a developed Knott model is proposed by introducing the interphase region for dielectric constant estimation of polymeric nanocomposites.

The article is well-written but, in my opinion, needs major revision before it can be accepted for publication in Polymers.

Q1) Eq.7 and 8:

Vp nomenclature is defined for both the volume and effective volume in Eq.7 and 8, respectively, please use another nomenclature for the volume of effective particle.

Q2) Eq. 10:

Please explain the differences of this equation with the one proposed by Tanaka (please refer to Fig.2b of Reference 36).

Q3) Page 4, line 155-156:

quote "... Introducing the interphase into Knott model is responsible for the accurate simulation results..."

Please add the results of modeling with the Knott model in Fig. 2 and 3 to visualize clearly the previous statment.

Q4) Figs. 2 & 3:

These figures must include experimental data of other samples to validate the model. For instance, the report of Kochetov et al (Reference 38) presents experimental data for SiO2-epoxy resin nanocomposites but more experimental data of other samples must be added.

Reviewer 2 Report

This is an interesting paper aiming at developing the general Knott model, taking into consideration the interphase region that affects the properties of the nanocomposites. The paper may be published in the Polymers journal, however, after some important corrections, as justified in the following points:

- It is noticed that the experimental results of the dielectric constant were retrieved from the published literatures. However, the authors should include some basic details about the preparation of polymer-based nanocomposites;

- Why only BaTiO3-epoxy resin and Al2O3-epoxy resin nanocomposites were selected? It could be explained in more detail;

- I understand that the proposed Knott model correlates well with the experimental data. Could the mathematical model be applied in order to predict the dielectric properties of carbon-based polymer nanocomposites? If yes, it would be interesting to verify the validity of the developed Knott model (including the effect of the interface) on a nanocomposite based on polymer matrix filled with carbon-based nanoparticles;

- Some of the important articles may be cited such as: The Journal of Chemical Physics 142 (2015) 194703; Polymer 149 (2018) 73-84; Polymer 203 (2020) 122785.

Round 2

Reviewer 1 Report

Authors have addressed my major concerns about the manuscript in the original form, and in my opinion, the revised version can be accepted for publication in "Polymers".